

# Ice nucleation by viruses and their potential for cloud glaciation

Michael P. Adams[1, #], Nina S. Atanasova[2, 3, #], Svetlana Sofieva[2, 3], Janne Ravantti[3], Aino Heikkinen[2 +], Zoé Brasseur[4], Jonathan Duplissy[4,5], Dennis H. Bamford[3,*] & Benjamin J. Murray[1*]

[1] - Institute for Climate and Atmospheric Science, School of Earth and Environment, University of Leeds, Leeds, UK
[2] - Finnish Meteorological Institute, Helsinki, Finland
[3] - Molecular and Integrative Biosciences Research Programme, Faculty of Biological and Environmental Sciences, University of Helsinki, Finland
[4] - Institute for Atmospheric and Earth System Research / Physics, Faculty of Science, University of Helsinki, Finland
[5] - Helsinki Institute of Physics, University of Helsinki, Helsinki, Finland
[#] Authors contributed equally to the work
[+] Now at the Institute for Molecular Medicine Finland, HiLIFE, University of Helsinki, Helsinki, Finland

*Correspondence to*: Dennis.Bamford@helsinki.fi and B.J.Murray@leeds.ac.uk

**Abstract.** In order to effectively predict the formation of ice in clouds we need to know which subsets of aerosol particles are effective at nucleating ice, how they are distributed and where they are from. A large proportion of ice-nucleating particles
(INPs) in many locations are likely of biological origin, and some INPs are extremely small being just tens of nanometers in size. The identity and sources of such INPs are not well characterized. Here, we show that several different types of virus particles can nucleate ice, with up to about one in twenty million virus particles able to nucleate ice at -20°C. In terms of the impact on cloud glaciation, the ice-nucleating ability (the fraction which are ice nucleation active as a function of temperature) taken together with typical virus particle concentrations in the atmosphere lead to the conclusion that virus particles make a
minor contribution to the atmospheric ice-nucleating particle population in the terrestrial influenced atmosphere. However, they cannot be ruled out as being important in the remote marine atmosphere. It is striking that virus particles have an ice-nucleating activity and further work should be done to explore other types of viruses for both their ice-nucleating potential and to understand the mechanism by which viruses nucleate ice.



# 1    Introduction

The formation of ice in clouds is critically important for the planet's radiative balance and our prediction of future changes in climate with increased greenhouse gas concentrations (Vergara-Temprado et al. 2018; Tan, Storelvmo, and Zelinka 2016). Ice-nucleating particles (INPs) have the potential to cause supercooled liquid cloud droplets, present in mixed-phase clouds, to freeze at temperatures greater than homogenous freezing, which can drastically alter cloud properties such as albedo, composition and lifetime (Murray et al. 2012; Hoose and Möhler 2012; Kanji et al. 2017). Despite the potential importance of INPs, there is still a lack of knowledge regarding their characteristics, sources, and ultimately their temporal and spatial distribution around the globe.

Our current knowledge of atmospheric INPs (under mixed-phase cloud conditions) suggests a number of potentially important aerosol types, including mineral dust, marine organics and terrestrial bioaerosols (DeMott et al., 2010; Kanji et al., 2017). The characteristics and source regions for mineral dust are relatively better understood than other potentially important INPs, and mineral dust from both high (Sanchez-Marroquin et al. 2020; Tobo et al. 2019) and low latitude sources is thought to be the dominant INP around much of the globe at temperatures $< -20\ °C$. Marine organics and terrestrial bioaerosols have both been demonstrated to play a major role in the global INP burden, but the nature of these INPs are less well understood than that of mineral dust. Marine organics are of particular importance in remote marine regions where there is little mineral dust (T. W. Wilson et al. 2015; Burrows et al. 2013). Terrestrial bioaerosols are thought to outcompete mineral dust in the terrestrial mid-latitudes at temperatures $> -20\ °C$, however their source(s) and nature are at present poorly understood (Conen et al., 2016; McCluskey et al., 2018; O'Sullivan et al., 2018; Vergara-Temprado et al., 2017).

Known INPs of biological origin include bacteria, fungi, pollen and marine organics amongst others (Kanji et al. 2017). Bacteria and fungi exhibit ice nucleation due to the presence of ice-nucleating proteins (Green and Warren 1985; Pouleur et al. 1992; Lindow', Arny, and Upper 1982), whilst the ice-nucleating ability of pollen has been linked to polysaccharides (Pummer et al. 2012; Dreischmeier et al. 2017). Marine organic INPs, associated with sea spray, are thought to be biogenic and are often smaller than 0.22 μm, but it is currently not clear exactly what these ice-nucleating particles are and there may be multiple marine INP types (Creamean et al., 2019; DeMott et al., 2016; Irish et al., 2017, 2019; Schnell et al., 1975; Wang et al., 2015; Wilson et al., 2015). Compared to non-biological INPs, some microorganisms such as specific bacteria or fungi nucleate ice at relatively high temperatures; for example, the best-studied ice nucleating bacterium, Pseudomonas syringae, can nucleate ice at temperatures up to −2 °C (Morris et al., 2004; Morris et al., 2013). Despite the ice nucleation potential of primary biological aerosol particles, recognized since 1970s (Schnell et al. 1976), the global distribution and sources of biological INPs remain poorly understood (Murray et al. 2012; Kanji et al. 2017). Hence characterizing the ice nucleating ability of the various categories of biological aerosol particles is important.

In bacteria, membrane proteins are thought to interact with water and impose order in supercooled water in such a way as to promote nucleation of ice. Pandey et al. (2016) demonstrated that in the case of P. syringae patterned hydrophilic-hydrophobic



regions due to the interactions of amino acids belonging to the membrane protein led to the increased ordering of water molecules coupled with efficient removal of thermal energy from the surrounding water molecules into the bacterial cell. This

mechanism could potentially protect microorganisms at sub-zero temperatures and preserve their viability and infectivity in the atmosphere (Wilson, Grogan, and Walker 2012; Morris, Monteil, and Berge 2013). Whether or not a bacterium has the potential to produce ice-nucleating proteins is dependent on the presence of an ice nucleation gene. At present, eight ice-nucleating proteins are known and reviewed in the protein database UniProt, each with an associated gene (protein IDs: O33479, P06620, Q47879, P16239, O30611, P09815, P20469, P18127). It is thought that a single functional ice nucleation

protein gene in bacteria is both necessary and sufficient for ice nucleation activity. The INA of a bacterium that has a gene for the ice-nucleating protein in its genome depends on the expression of the gene (i.e., if the protein coded by the gene is actually produced by the bacterium), the integration of the protein into the outer membrane of the bacterial cell and stabilization of the protein complex by the surrounding membrane constituents.

Viruses are a presently under-studied source with respect to their potential as atmospheric biological INPs. Very little is known

about viruses in the atmosphere in general, and even less about their potential to influence cloud properties through cloud glaciation. The only studies we are aware of in which the ice-nucleating ability of a virus was examined was that of Junge & Swanson (2008) (who studied the polar Colwellia phage virus) and Cascajo-Castresana et al. (2020) who studied a series of common proteins and a single virus. The former found that these virus particles did not nucleate ice in their experimental system. The latter observed ice nucleation activity in the Tobacco mosaic virus (TMV), a plant virus that infects the family of

Solanaceae such us tobacco, tomato, or pepper. TMV was shown to be above the baseline of the buffer solution it was suspended in, and it was noted in the study that whilst TMV had a lower onset freezing temperature of other samples in the study (a range of proteins), when normalized to cumulative active site density it was more active.

Compared to bacteria and other micron-sized, single-celled microorganisms, viruses are considerably smaller (from ~25 nm in diameter; except for the nucleocytoplasmic large DNA viruses that are cellular size). The small size of virus particles means

that their atmospheric lifetime has the potential to be on the order of many days to weeks in the atmosphere, although this will depend on the size of the particles that they are internally mixed with. This is considerably longer than the lifetime of larger biological particles, especially those larger than ~10 μm, which have lifetimes of only hours (Grythe et al. 2014; Reche et al. 2018) and therefore have atmospheric abundances which decrease rapidly during transport (Hoose, Kristjánsson, and Burrows 2010).

In addition to supermicron entities such as bacteria, submicron sized biological particles have also been shown to be effective ice nucleating particles (O'Sullivan et al. 2015). For example, it has been shown that there are biological INPs belonging to fungal and pollen samples at sizes below 200 nm (Pummer et al. 2012; Fröhlich-Nowoisky et al. 2015). Fertile soil samples when dispersed in water and filtered, have also been shown to have a significant number of ice-nucleating particles below 200 nm (O'Sullivan et al. 2015; Hill et al. 2016). O'Sullivan et al. (2015) showed that some ice nucleation persisted in fertile soil



samples filtered to 1000 kDa, however ice nucleation above −10 °C was removed by these filtrations. Decayed plant litter was shown to have comparable INP concentrations before and after filtration through 200 nm filter pores, and retained a fraction of these INPs when further filtered through 20 nm filter pores (Vali et al. 1976). Ice-nucleating particles below 200 nm were measured in North American Arctic snow samples and in precipitation from North China temperate grassland (Du et al. 2017; Rangel-Alvarado, Nazarenko, and Ariya 2015). The snow samples were shown to be of biological origin and subsequently

tested for virus-like structures, of which none was observed. Despite this, the authors stated they could not preclude viruses as a potential explanation for the observed ice-nucleating activity, based on the size of the INPs and their likely origin. Measurements of INPs in the Arctic sea surface microlayer showed that most of the observed ice nucleation (in the immersion mode) was caused by particles between 0.02 and 0.2 µm in size and were heat labile; viruses were suggested as a potential explanation (Irish et al. 2017; T. W. Wilson et al. 2015). Atmospheric measurements made in the Arctic showed the presence

of atmospheric INPs in the size range 150 - 340 nm (Creamean et al., 2019; Creamean et al., 2018). Size resolved measurements made in a boreal forest in Hyytiälä, Finland showed an instance in which INPs in the size range 250 - 500 nm dominated the atmospheric INP burden at temperatures > −22 °C, whilst measurements made at near surface level locations in the U.K showed INPs present at sizes below 250 nm (Porter et al. 2020). There is a growing body of evidence that suggests there is a reservoir of currently unidentified biological particles in the fine mode (<250 nm) present in soil/plant life, the oceans and the

atmosphere. In this study, we test the hypothesis that viruses are a potential candidate for the source of these fine mode INPs.

It has been estimated that there are ~$10^{31}$ virus particles in the biosphere (Whitman, Coleman, and Wiebe 1998), with approximately $10^7$ virus particles per ml of seawater, $10^8$-$10^9$ per ml in marine surface sediments (Suttle 2007; 2005) and $10^8$-$10^9$ per gram of soil in different types of terrestrial environments (Srinivasiah et al. 2008). Numerous studies indicate that there are approximately 10-100 times more viruses compared to their host cells in any given environment (Srinivasiah et al. 2008;

Cai et al. 2019). With respect to viral abundance in the atmosphere, there is at present a dearth of knowledge. Rastelli et al. (2017) measured the viral abundance in both the seawater microlayer and the aerosol phase directly above using a bubble generator system designed to mimic wave breaking in open seawater. Virus concentrations for seawater and sampled air were $5 \times 10^{11}$ virus particles m$^{-3}$ and 0.3–$3.5 \times 10^5$ virus particles m$^{-3}$, respectively. Virus particles were measured from outdoor air samples using a filter based technique taken at a University campus, with the atmospheric virus particle concentration being

measured as $1.2 \pm 0.7 \times 10^6$ virus particles m$^{-3}$ (Prussin, Garcia, and Marr 2015). The spatial and temporal variability of airborne viruses were investigated in a series of different locations (residential district, forest and an industrial complex), with concentrations of $1.7 \times 10^6$ to $4.0 \times 10^7$ virus particles m$^{-3}$ being measured (Whon et al. 2012). Overall, the range of outdoor virus concentrations recorded in the literature range between $0.3 \times 10^5$ to $4.0 \times 10^7$ virus particles m$^{-3}$. It is likely that these numbers do not represent the full variability of virus particle concentration due to the scarcity of measurements.

Despite the large number of virus particles measured in various environments there are a relatively small number of different particle structures a virion (an infective virus particle) can have. This is due to physical constrains of protein fold space that make up the virus particle architecture (Abrescia et al. 2012). Structurally similar viruses can have different host organisms





and different geographical source locations (Bamford 2003; Saren et al. 2005; Atanasova et al. 2012) . There are several observations of virus isolates with high genome identity originating from spatially distant environments (Atanasova et al. 2015;
Pietila et al. 2012; Saren et al. 2005; Tschitschko et al. 2015). We have chosen virus particles for this study that represent several different symmetric or asymmetric virus architecture types, icosahedral, icosahedral with internal lipid membrane, icosahedral enveloped and lemon-shaped. As it would be beyond the realms of feasibility to test even a minute fraction of the 10³¹ different viruses in the biosphere, we took the approach that we believe allows us to investigate the maximum parameter space and test the hypothesis that virus architecture/structure controls ice-nucleating ability. In this study we present the ice-
nucleating ability of viruses with these different architecture types, demonstrate the potential of different structural components in viruses to nucleate ice, and attempt to estimate the potential of viruses as a class of atmospheric ice-nucleating particles.

## 2    Methods

### 2.1    Virus growth, purification and production of Phi6 subviral particles

Virus particle suspensions were produced under carefully controlled conditions which resulted in suspensions of high purity;
the details of this preparation are given here. Bacterial and archaeal strains and viruses used in this study are listed in Table S1. Bacterial host strains were aerobically grown in Luria-Berthani broth at 28 °C for *Pseudomonas syringae* pathovar *phaseolicola* HB10Y, and *P. syringae* LM2489, and at 37 °C for *Escherichia coli* HMS174 and *E. coli* C122 strains. Archaeal host strains were aerobically grown in 23 % Modified Growth Medium (MGM) at 37 °C (Nuttall and Smith 1993).

Here, 1× signifies once purified particles and 2× twice purified, concentrated virus sample. Bacteriophages PRD1 and Phi6
were 1× purified as described in Bamford et al. (1995). The 2× purification of PRD1 was performed (Bamford et al. 1995; Lampi et al. 2018). The PRD1 particles devoid of DNA (procapsids) were collected after 1× purification, during which the DNA containing particles sediment further along the sucrose gradient compared to the empty procapsids. The 1× purified Phi6 was further purified to 2× by density gradient ultracentrifugation in 20-70 % sucrose in 20 mM K-phosphate buffer pH 7.2 with 1 mM MgCl₂ (designated here as K-phosphate buffer) followed by concentration as described in Bamford et al. (1995).
Viruses Phi8, Phi12, Phi13 and, Phi2954 were produced and precipitated according to Qiao et al. (2010), and the 1× purification was performed by rate-zonal ultracentrifugation in 5-20 % sucrose gradients in K-phosphate buffer, Sorvall AH629 rotor, 24 000 rpm, 50 min, 15 °C, followed by concentration using differential ultracentrifugation, Sorvall T865 rotor, 34 000 rpm, 3 h, 10 °C. All other viruses were purified to 1× preparations according to protocols described in Eskelin et al. (2019) (for PhiX174), Pietilä et al. (2009) (for HRPV-1), Pietila et al. (2012) (for HRPV-6), Demina et al. (2016) (for HCIV-1) and Bath
et al. (2006) (for His1).

Phi6 subviral particles were prepared according to Bamford et al. (1995), modified by Eskelin & Poranen (2018) (for butylated hydroxytoluene treated particles). Phi6 NC were prepared by adding 1 % final concentration of Triton X100 to 1× purified Phi6 particles in K-phosphate buffer and incubating 30 min at 22 °C. The treated particles were collected by ultracentrifuge,





Ti1270 rotor, 30 000 rpm, 4 h, 15 °C. Particles were flushed three times with and resuspended in 0.5 ml K-phosphate buffer
overnight at 5 °C.

The protein concentration of viral and subviral particles was measured by the Bradford assay using bovine serum albumin as
a standard (Bradford 1976). Virus samples were analyzed by sodium dodecyl sulfate- 16% polyacrylamide gel electrophoresis
(SDS-PAGE) (Olkkonen and Bamford 1989) to visualize viral protein profiles.

## 2.2    Search for ice nucleation motifs

Currently, there are eight referenced ice nucleation proteins identified from bacterial cells according to the public protein
database (UniProt, https://www.uniprot.org/). The ice nucleation motifs (INMs) predicted based on these genes are short
protein sequences conserved in this protein family. They are abundant for the ice nucleation proteins (IN proteins), but scarce
in the rest of the bacterial genomes. The group of motifs specific for a protein family can serve as a functional fingerprint
indicating similarities in structure and function. It was previously determined that INM3 corresponds to the clathrate structure
part of the protein responsible for ice nucleation activity in bacterial IN proteins (Gurian-Sherman and Lindow 1993; Kajava
and Lindow 1993).

The INMs were acquired from SPRINT, an interface for PRINTS data bank of protein family fingerprints. SPRINT is a public
domain database currently maintained at the University of Manchester (http://130.88.97.239/dbbrowser/sprint/). The
INMs can be found in SPRINT by identifier ICENUCLEATN. All known ice nucleation motifs in IUPAC (International Union
of Pure and Applied Chemistry) nomenclature are listed in Table S2. Since some of the putative viral proteins are not fully
characterized, we used protein INMs from SPRINT to build generalized nucleotide motifs. The annotated viral genomes were
acquired from the National Centre for Biotechnology Information (NCBI) genome database (Table S3).

Ice nucleation motifs were searched for in the viral genomes using MEME (Multiple Em for Motif Elicitation) Suite 5.1.0.
(Bailey et al. 2009). The search was performed using MCAST (Motif Cluster Alignment Search Tool) and FIMO (Find
Individual Motif Occurrences)-tools (Grant et al., 2011; Bailey & Noble, 2003). MCAST searches for input motifs in the query
sequence for statistically significant clusters of non-overlapping occurrences. FIMO, in turn, searches for individual motif
occurrences in the sequences, each motif independently. Each found occurrence was scored with p-value. The p-score
thresholds for significant findings were set to 0.0001.

Putative IN-proteins were determined in the viral genomes based on the repetitiveness of IN motif occurrences in the
sequences, as well as the total INM coverage. The INM coverage is calculated from the total length of the protein sequence
matching the INM sequences compared to the total length of the protein. The INMs were annotated to the sequences using
Artemis 17.0.1 and the protein alignments were performed using Muscle 3.8.425 and visualized using Geneious Prime
2020.1.1. All the potential IN proteins are listed in Table 1.



## 2.3   Ice nucleation experiments

Samples for analysis of the ice nucleating activity of virus particles were prepared by diluting 1× or 2× purified virus particles to specific buffer solutions (Table S1) so that the final concentration of pfu/ml was $10^{10}$-$10^{12}$. Subviral particles were used without dilution. Virus host strains were collected by centrifugation (Eppendorf, 13 000 rpm, 5 min, 22 °C), diluted into the same buffer as the virus (Table S1), centrifuged (Eppendorf, 13 000 rpm, 5 min, 22 °C) and resuspended into buffer according to Table S1.

Ice nucleation experiments were carried out using the µl-NIPI (nuclation by immersed particles instrument) (Whale et al. 2015). In brief, the µL-NIPI analysis involved pipetting 1 µL droplets (in which sample material was suspended) onto a hydrophobic-coated glass cover slip atop an aluminium cold stage, and cooled. Viral samples were vortexed for 30 seconds prior to being pipetted to ensure the particles were evenly distributed through the suspension. Droplet freezing was recorded using a camera, with the freezing temperature of each droplet recorded.

The cumulative fraction of droplets frozen on cooling to a temperature, $f_{\text{ice}}(T)$ is defined by:

$$f_{\text{ice}}(T) = \frac{n_{\text{ice}}(T)}{N_{\text{tot}}} \tag{1}$$

Where $n_{\text{ice}}(T)$ is the cumulative number of droplets frozen on cooling to $T$ and $N_{\text{tot}}$ is the total number of droplets. The cumulative number of active sites per particle, $n_n(T)$, was calculated according to equation 2:

$$n_n(T) = \frac{-ln(1-f_{\text{ice}}(T))}{n_{\text{v}}} \tag{2}$$

Where $n_{\text{v}}$ is the number of virus particles per 1 µl droplet. The cumulative active sites per unit mass of material, $n_{\text{m}}(T)$ is defined by:

$$n_{\text{m}}(T) = \frac{-ln(1-f_{\text{ice}}(T))}{m_{\text{v}}} \tag{3}$$

Where $m_{\text{v}}$ is the mass of virus particles per droplet.

The freezing point depression of pure water due to NaCl (i.e. in the buffer solutions) was calculated using:

$$\Delta TF = K_F \cdot b \cdot i \tag{4}$$

where $\Delta T_F$ is the freezing depression, $K_F$ is the cryoscopic constant (1.853 K kg mol$^{-1}$ for water), $b$ molality and $i$ is the Van't Hoff factor (2 for NaCl). Hence, we report the degree of supercooling relative to the melting point of the aqueous saline solution. The correction was typically about 1°C for most virus suspensions (it was around 3°C for an Archea virus which required a very high salt concentration).



## 3    Results

### 3.1    Ice nucleating ability of virus particles

We studied virus ice nucleation from a virus structural perspective using the nucleation by immersed particle instrument (μl-NIPI) technique (Whale et al. 2015). We examined the ice nucleation activity (INA) of 11 viruses with different particle architectures, in an effort to probe the hypothesis that virus architecture/structure influences the ice-nucleating ability of virus particles. (Figure 1). These viruses included five enveloped Cystoviruses of *P. syringae* hosts with particle diameters of ~85 nm (Phi8, 6, 12, 12 and 2954; Figure 1A), two icosahedral viruses with an internal lipid membrane and particle diameters of ~70 nm (Figure 1B, PRD1 and HCIV-1), one of the icosahedral viruses without the DNA (Figure 1B, PRD1 no DNA), one 30 nm icosahedral virus without lipids (Figure 1B, PhiX174), two enveloped pleomorphic viruses with particle diameter of ~50 nm (Figure 1C), and one lemon-shaped virus (Figure 1D). Phi6-like viruses are commonly used as models for viruses that cause respiratory illnesses like SARS-CoV-2, the causative agent of COVID-19, due to structure similarity. Of the eleven viruses tested, nine showed an INA distinct from the INA of the buffer solution they were suspended in (Figure 1E and S2).

Phi12, an enveloped virus infecting *Pseudomonas syringae*, was found to be the most ice nucleation active virus in our study (in terms of the number of INPs per virus particle, $n_n$). Phi12 was observed to trigger freezing from −15°C to −21°C, with $n_n$ values between $5\times10^{-10}$ to $5\times10^{-8}$ particles$^{-1}$. The other structurally similar cystoviruses of *P. syringae* were all ice nucleation active, although less so compared to Phi12 (Figure 1E). At this point we address the question of if the host bacterial cells might introduce ice nucleating entities which might be confused with the virus particles. In particular, several ice nucleation active *P. syringae* strains have been described in previous studies (de Araujo et al. 2019). However, none of the *P. syringae* host strains of the viruses used in this study exhibited an INA distinguishable from that of the buffer solution (Table S1, Figure S1a). Also, none of the strains contain a functional *ina*-gene, only partial pseudogenes. This indicates that the INA observed here is solely due to the virus particles. In addition (and more importantly), the virus samples used in the μl-NIPI test were purified according to established virus purification protocols (See Methods) and the purity was verified on sodium dodecyl sulfate-polyacrylamid gel electrophoresis (SDS-PAGE) (Figure S2). Furthermore, a well-established model virus, Phi6, was purified twice and the results of both purifications were examined on the μl-NIPI assay strongly implying that the virus particles are responsible for the observed INA (Figure S3) rather than contamination from the process or host bacteria.

The source of the INA was further studied using two of the best-characterized model viruses, Phi6 of *P. syringae*, and PRD1 of *E. coli*. Regarding Phi6, we used biochemical dissociation to disassemble the virus particles into sub-viral particles (Figure 2). First, the virus spike proteins were removed using butylated hydroxy toluene (BHT), with the resulting particle referred to as Phi6 BHT and the separate spike proteins referred to as Phi6 P3. Secondly, the lipid envelope and the associated proteins were removed using the anionic detergent Triton X100, exposing the nucleocapsid (NC) structure of Phi6 virion (Figure 2A). Each of the sub-viral particles was shown to have an INA distinguishable from the K-phosphate buffer (Figure 2B). Each of the sub-viral components, along with Phi6, was normalized to the mass of particles per volume of sample ($n_m$). When



normalized in this manner, each component spanned approximately the same range in $n_m$ space, $10^2 - 10^4$ (mg$^{-1}$) across a range of temperatures. The freezing spectrum of each component was similar, with the Phi6 BHT sub-viral components having slightly warmer freezing temperatures than Phi6 at equivalent $n_m$ values, whereas spike proteins (P3 in Figure 2A) had a slightly

lower freezing temperature than Phi6. NC was found to be the most IN active sub-viral particle of Phi6 (Figure 2C), freezing approximately 4 °C warmer across the measured $n_m$ range when compared to Phi6. INA is in part related to size (Pummer et al. 2015), hence since the spike proteins are only ~20 nm in diameter, whereas the BHT and NC particles are close to 80 nm. Hence, the difference in activity may be related to size. It is not clear how virus particles behave in the atmosphere, but several environmental stressors can disrupt virus particles exposing their internal parts. The other Phi6 sub-viral particles were also

IN active (Figure 2) indicating, that the virus has broad IN potential, either being active as a whole or in a disrupted form. PRD1 was measured for its INA both with and without DNA. The $n_n$ values for PRD1 with and without DNA are shown in Figure 3, and are similar to one another. This result suggests that the presence of PRD1's DNA is not related to the INA of the particles.

To further our understanding of the influence of virus structure on IN activity, we tested six other viruses, four archaeal and

two bacterial. Of these six viruses, two archaeal viruses (HRPV1 and HRPV6) were enveloped like Phi6, but lack particle symmetry and an NC structure (Figure 1C). HRPV1 (Figure S1b) was not distinguishable from the Saline buffer (Table S1) it was suspended in, whilst HRPV6 (Figure S1b) was distinguishable from the buffer, but was not distinguishable from its host, and as such are shown as limiting values (Figure S5). Viruses with icosahedral symmetry that contain an internal lipid membrane (PRD1 and HCIV-1, Figure 1B) were also tested to further probe the dependency of viral INA on structure. PRD1,

a well-known model virus (Bamford et al. 1995) was shown to be INA with a signal distinguishable from both the K-phosphate buffer and its host (Figure S6), and $n_n$ values comparable to that of the majority of the *P. syringae* viruses (excluding Phi12) (Figure 1E). HCIV-1 did not have an INA distinguishable from the Saline buffer it was suspended in and is thus shown as a limiting value (Figure 1B). Another icosahedral virus, PhiX174, this time without a lipid membrane (Figure 1B) was tested and had $n_n$ values similar to that of PRD1 and the majority of the *P. syringae* viruses. We further studied the INA dependency

on virus architecture by studying an asymmetrical lemon-shaped archaeal virus, His1 (Bath et al. 2006). Interestingly, the virus had a higher INA than all the tested viruses, except for Phi12 (Figure 1D-E), indicating that structurally different viruses, symmetric or asymmetric, can be IN active. His1 was shown to be distinguishable from the saline buffer solution (Figure S1b) and its host (Figure S7).

### 3.2    Genetic analysis of ice active virus particles

The genomes of the 11 viruses included in this study were explored by bioinformatic analysis to further examine the source of IN activity. The ice nucleation activity observed in bacteria is due to protein structures which mimic ice crystal clathrate structure on the cell surface thus facilitating ice crystal formation around the cell (Kajava and Lindow 1993). In viruses, the source of INA might also be manifested due to proteinaceous origin. The possibility of the capsid or membrane proteins in



virus particles possessing similar structure and function as explanation for their ice nucleation capacity was explored in this
study. This hypothesis was approached using ice nucleation motifs, specific and conserved short sequences in IN proteins, to
search for the proteins potentially capable of nucleating ice.

Viral proteins with significant INM coverage and presence of INM3 in their sequence were predicted in eight of the viruses
(Table 1). According to the results, only Phi13 and PhiX174 did not have potential IN proteins, with His1 having coverage
below 15% and so is not shown in the table. Other viruses contained at least one potential protein with INM coverage of 15-
50% and obligatory INM3 presence. However, the INA of Phi13 and PhiX174 is similar to the majority of the tested viruses
such as Phi6. Similarly, HRPV1 and HCIV1 contain potential IN proteins, but these viruses had the weakest INA of the tested
virus particles. Therefore, the presence of INMs in the sequence does not correlate to the capacity to nucleate ice.

## 4    Implications for the atmospheric ice nucleating particle population

In order to estimate the INP concentrations associated with virus particles in the atmosphere we have combined the $n_n$ values
shown in Fig. 1E and the upper limit of the concentration of viruses in the atmosphere from literature data (taken as $4 \times 10^7$
particles m$^{-3}$; see discussion in the introduction). It is important to note that we based these virus INP concentrations on the
INA of the specific samples which we studied and it may be possible that other virus particles have greater INA. However,
since there are a limited number of virus architectures and we test a range of these architectures, we tentatively suggest that
that we capture the typical range of INA of virus particles.  Also shown on Figure 4 are envelopes showing the range of data
from field campaigns in terrestrial (orange) and remote marine/polar environments (green) (see Table S4 for a list of
representative measurements included in these envelopes).

As discussed in the introduction, in terrestrial environments, mineral dust is thought to be a very important INP type, with
marine organics playing a secondary role (Vergara-Temprado 2017). In addition, there is evidence that biological INP play an
important role in the terrestrial mid-latitudes (O'Sullivan et al. 2018; Hill et al. 2016; Šantl-Temkiv et al. 2019; Conen, Stopelli,
and Zimmermann 2016; Pratt et al. 2009). Figure 4 shows that across the entire temperature spectra relevant for mixed-phase
clouds, the concentration of virus INPs are lower than the lowest typical INP concentrations in terrestrial influenced areas. The
closest the terrestrial envelope and the virus data points come to overlapping is between −18 °C to −22 °C, at which
temperatures the difference in INP concentration is approximately one order of magnitude, i.e. at most they might contribute
about 10% of the INP population at around -20 °C. Furthermore, one might expect that in environments where there is a strong
source of virus particles, there is also a strong source of other biological materials, hence the influence of virus INP may be
overestimated in our simple analysis. Overall, these results suggest that virus INPs generally play a minor role in regions
influenced by terrestrial INPs.

Remote marine locations are less influenced by active terrestrial sources and thus the INP populations there are different from
those of the terrestrial atmosphere (Creamean et al., 2019; DeMott et al., 2016; McCluskey et al., 2018a; McCluskey et al.,





2018b). Marine organics and sea spray aerosol have been shown to be INP sources of first order importance in such environments (DeMott et al., 2016; Vergara-Temprado et al., 2017; Wilson et al., 2015). There have been field measurements made in remote marine environments, which have reported remarkably low INP concentrations. McCluskey et al. (2018b) measured INP concentrations in a pristine marine environment at the Mace Head research station in 2015, with INP concentrations as low as $10^{-3}$ L$^{-1}$ at −20 °C. In a separate field campaign, measurements were made in the Southern Ocean,

INP concentrations range between $3.8 \times 10^{-4}$ to $4.6 \times 10^{-3}$ at −20 °C (McCluskey et al., 2018a). Figure 4 shows overlap between the virus INP data points and the marine envelope in the temperature range −15 °C to −27 °C, with the most active of the virus INPs being approximately 15× higher than the lower limit of the marine envelope at −20 °C. Whilst this by no means proves that virus INPs are important in remote marine environment, it indicates they may contribute to the atmospheric INP burden in such regions. However, the lowest INP concentrations in the remote marine environment are most likely associated with

periods when the aerosol concentrations were lowest, as a result of the combined effect of precipitation scavenging and weak sources. Under these conditions, virus particles would also presumably be depleted.

## 5   Summary and conclusions

In this study we show a range of viruses can nucleate ice heterogeneously when immersed in supercooled solution droplets. A selection of virus types with diverse architectures are shown to have ice-nucleating abilities spanning three orders of magnitude

at −20 °C, when normalized to particle number. We probed the virus ice-nucleating ability dependence on virus particle structure/architecture, showing that for our selected viruses there was not a dependency on virus architecture. Bioinformatic analysis shows that our current knowledge of ice nucleation due to *ina* genes/proteins exhibited by bacterial ice nucleators is likely insufficient to understand why viruses nucleate ice, which can be due to e.g. the overall arrangement of structural proteins making up the virus particles.

Our results are based on a small subsample of virus types but include several of the most prominent viral architectures. Nine out of 11 tested viruses were ice nucleation active indicating that several structurally different viruses can have IN potential. In addition, it has been shown previously that a tobacco mosaic virus can also nucleate ice (Cascajo-Castresana et al. 2020). While we have selected virus particles with a range of architectures that are relatively common in nature, it is possible that other virus particles nucleate ice more or less effectively. In particular, the specific virus types we have studied here are from

the terrestrial and freshwater aquatic environment; the isolation and testing of a range of marine viruses presents an important next step in quantifying the importance of viral ice nucleators. More work needs to be done to understand what drives viral ice nucleation, whether it would be dependent on virus structure/morphology, host, or some other factor.

This study shows the potential role viruses play as atmospheric INPs in certain environments. The ubiquity of viruses in the atmosphere implies they could serve as a baseline of INPs in situations where other, better-known atmospheric INPs are absent

in any meaningful quantity. However, our estimates for the upper limit of virus INPs suggest they do not play a meaningful

role in terrestrial environment, but may contribute to the INP population in marine environments. More work needs to be done to understand both why viruses nucleate ice and what role they play in both regional and global atmospheric ice nucleation.

## Acknowledgements

Helin Veskiväli and Emeline Vidal are thanked for technical assistance. We thank Dr. Leonard Mindich for providing bacteriophages Phi12, Phi13, Phi2954 and *P. syringae* bacterial strain. Professor Ben Fane is thanked for providing PhiX174 bacteriophage. The use of the facilities and expertise of the Instruct-HiLIFE Biocomplex unit, member of Biocenter Finland and Instruct-FI, is gratefully acknowledged. N.S.A. wants to acknowledge the Academy of Finland Postdoctoral Grant 309570 and the Scientific Advisory Board for Defence grant VN/627/2020-PLM-9. We thank the European Research Council (648661
MarineIce, and 713664 CryoProtect) and the Natural Environment Research Council (NE/T00648X/1, M-Phase) for funding.

## Data availability

The data associated with this paper are openly available from the University of Leeds Data Repository https://doi.org/XXXXX.

## 350 Competing interests

The authors declare no competing interests.

## Author Contributions

M.A., N.S.A., S.S., A.H. and Z.B. performed the experiments. S.S. designed and performed bioinformatic analyses. D.B.
developed the NC purification protocol. M.A., N.S.A., B.M., and D.B. designed the experiments. All authors participated in data analysis and interpretation of results. N.S.A., J.D., J.R., B.M., and D.B. supervised and supported the project. The manuscript was written by M.A., N.S.A., S.S., B.M., D.B. All authors reviewed and approved the manuscript.

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




Figures and Tables

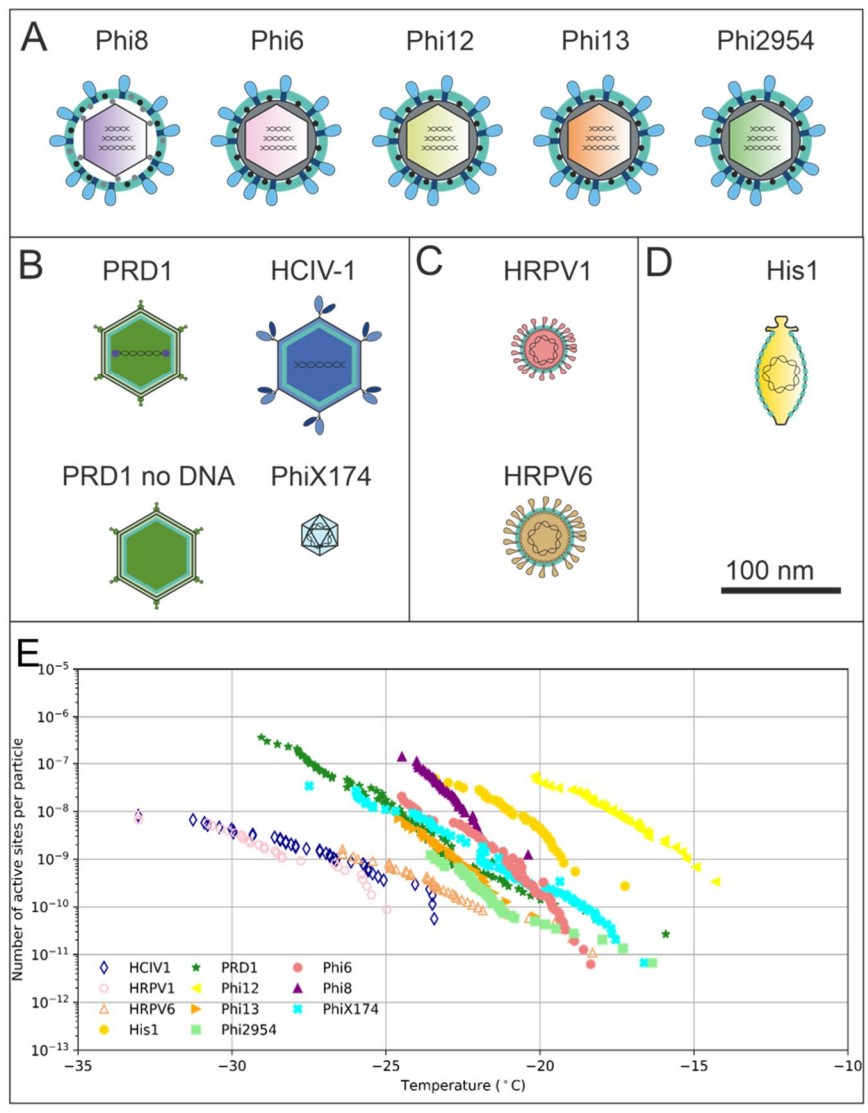

**Figure 1. Graphical representation of the virus particles used in the ice nucleation study and their ice nucleating ability. A) Enveloped icosahedral viruses. B) Icosahedral viruses. C) Pleomorphic viruses. D) Lemon-shaped virus. E) Ice nucleation activity plots (expressed as $n_n$), where hollow marker indicate limit of detection (LoD) measurements in which the freezing temperatures were consistent with the virus free saline buffer control. Virus particles are to scale according to the 100 nm scale bar. Temperature values**
**have been corrected for freezing point depression of NaCl.**



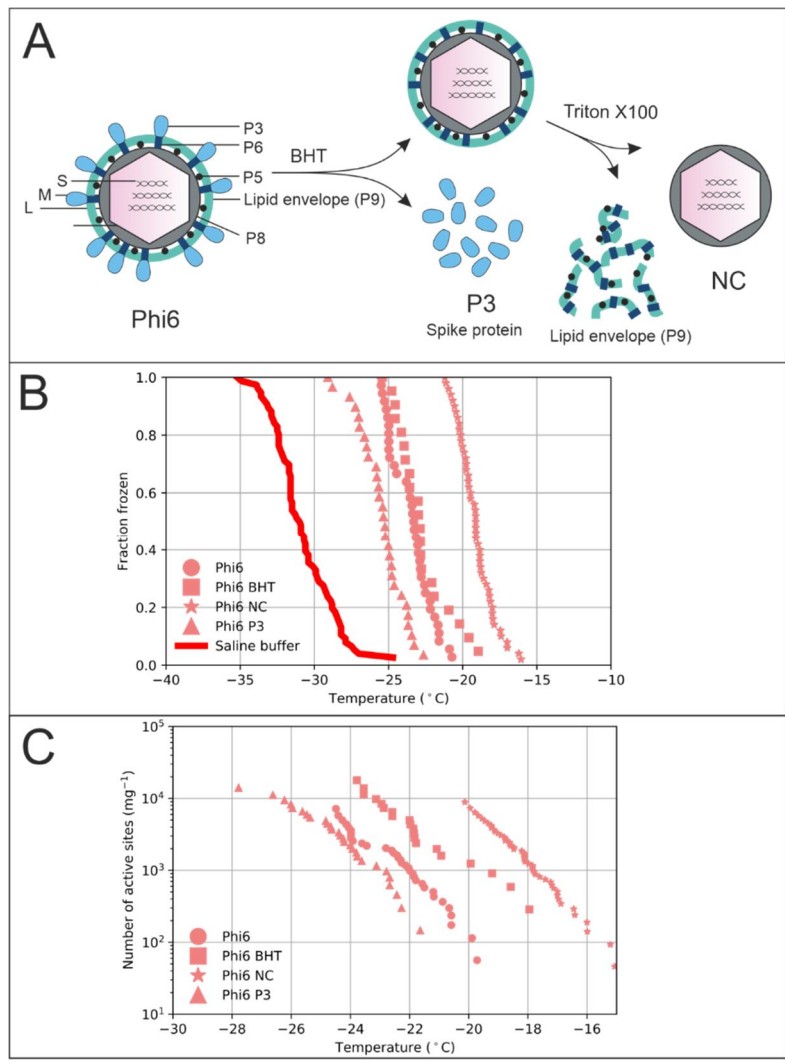

**Figure 2. Ice nucleation activity of the subviral particles of Phi6 virus. A) Biochemical dissociation of Phi6 virion. Small genome fragment is marked as S, medium genome fragment as M and large genome frgment as L; P3 is spike protein, P5 is lytic enzyme, P6 is membrane fusion protein, P8 is outer capsid lattice protein and P9 is major envelope protein; BHT means butylated hydroxyl toluene; NC is nucleocapsid. B) Fraction frozen curves for Phi6 and its sub-viral components. These values have not been correct for freezing point depression due to NaCl. C) The INA (expressed as active sites per unit mass, $n_m$) of Phi6 and its sub-viral components normalized to the mass of particle per volume of suspension. Phi6 BHT in graphs B and C refers to spikeless enveloped icosahedral structure. Temperature values have been corrected for freezing point depression of NaCl.**





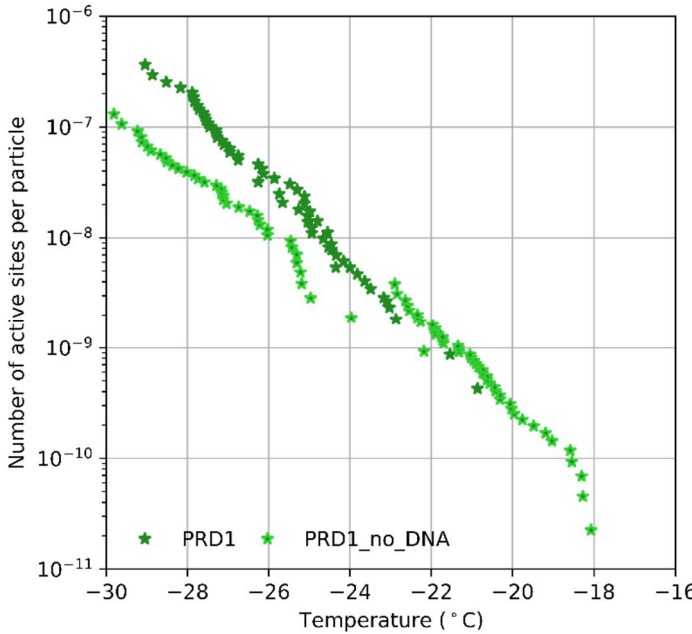

**Figure 3: The number of active sites per particle ($n_n$) for PRD1 with and without DNA. Temperature values have been corrected for freezing point depression of NaCl.**





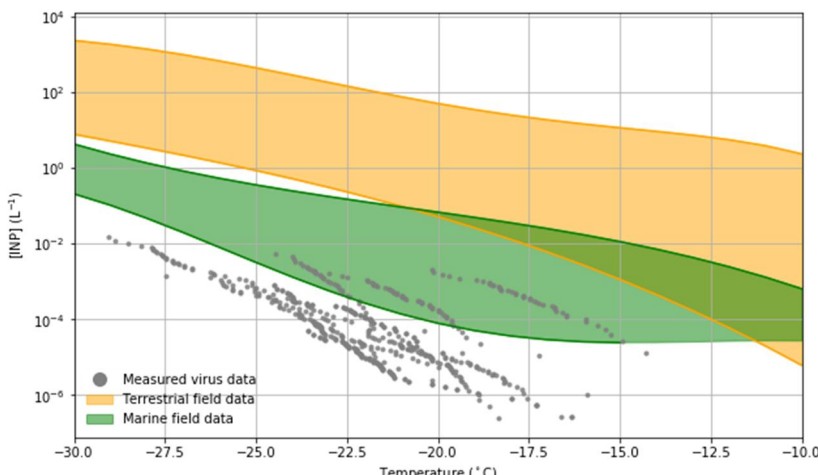

**Figure 4. Estimated viral INP concentration based on measured ice-nucleating ability of virus particles and upper limit literature**
**values of viral particle concentrations in the atmosphere compared to measured INP concentrations in both terrestrial (orange) and marine/polar (green) environments. Table S4 shows a list of the studies from which the data to create the field measurement envelopes were obtained. Temperature values have been corrected for freezing point depression of NaCl.**



**Table 1. Potential IN proteins, their location, function and INM-coverage.**

| Virus name | IN-protein candidates | Protein function | Location in virus | IN-motif coverage (%) | Protein ID number |
|---|---|---|---|---|---|
| **Phi6** | P3 | Adsorption to host cells, attachment to type IV pilus | spikes | 18 | NP_620351.1 |
| | P1 | Major inner capsid protein, RNA binding, replication and transcription | capsid | 20 | NP_620348.1 |
| **Phi12** | P3c | Putative host attachment protein | spikes | 16 | NP_690834.1 |
| | P9 | Membrane protein | external lipid membrane | 28 | NP_690828.1 |
| **Phi13** | P3b | Putative host attachment protein | spikes | 31 | NP_690814.1 |
| | P4 | Hexameric packaging NTPase | 5-fold vertices of the procapsid | 21 | NP_690818.1 |
| **Phi2954** | P3 | Host attachment protein | spikes | 18 | YP_002600769.1 |
| **PhiX174** | P11 | Minor spike protein | spikes | 29 | NP_040713.1 |
| **HCIV1** | Putative protein VP18 | Putative minor capsid protein | unknown hypothetically capsid | 24 | YP_009272867.1 |
| | VP3 | Capsid protein | capsid | 31 | YP_009272848.1 |
| **HRPV6** | ORF2 | Unknown | unknown hypothetically capsid | 33 | YP_005454286.1 |
| | ORF7 | Integral component of the membrane | external lipid membrane | 47 | YP_005454291.1 |



|  | ORF8 | ATP-binding, AAA-type ATPase | unknown hypothetically capsid | 16 | YP_005454292.1 |
|---|---|---|---|---|---|
| **PRD1** | P7 | Transclycosylase | lipid membrane | 18 | YP_009639979.1 |
|  | P14 | DNA delivery | lipid membrane | 31 | YP_009639980.1 |
