# Peer review of "Ice nucleation by viruses and their potential for cloud glaciation"

_Biogeosciences, 2020_

## Author Comment (AC1)

**COMMENTS TO REFEREE 1**

Review of „Ice nucleation by viruses and their potential for cloud glaciation" by Adams et al.

In the study, a range of different types of viruses are examined (partially also in modified form) concerning their ice nucleation activity and their possible contribution to ice nucleating particles (INP) on an atmospheric level. This has been a severely understudied field so far, and the study adds important knowledge to the field. Although the results "only" show, that viruses might not be a major contributor to these INP this is an important contribution to our understanding of biogenic INP.

It is a very well written and thoroughly conducted study. The structure of its former submission to a higher ranking journal still can be seen, but this does not make it less valid. It is great that this so far underexamined class of biogenic microorganisms, namely viruses, was examined with respect to their ability to nucleate ice with such great care, thought and thoroughness.

I only have a few minor / technical issues and can, after these have been corrected, fully support publication in Biogeosciences.

Minor issues:

line 76: "of other samples" -> "than other samples"

Reply: corrected.

lines 179 - 181: Maybe it has to do with my background and field of expertise, but I do not understand this part. Could you please put it in context a bit more? It concerns the following two sentences:

"Putative IN-proteins were determined in the viral genomes based on the repetitiveness of IN motif occurrences in the sequences, as well as the total INM coverage. The INM coverage is calculated from the total length of the protein sequence matching the INM sequences compared to the total length of the protein."

Reply: Lines 179-181 were adjusted (and simplified) accordingly: "In order to predict putative IN-proteins in the viral genomes, total INM coverage as well as the occurrence of repetitive IN motives was studied. INM coverage is calculated from the length of the matching INM sequence compared to the total length of the protein.".

line 185: For the non-biologists amongst your readers, explain "pfu".

Reply: Explanation added.

Also (as this becomes important in equation (2): How was this concentration of pfu determined?

Reply: The explanation "One plaque corresponds to the progeny of one virus that initially infected the host cell. Plaque forming units measures the number of infective virus particles in the sample." was added after the explanation of pfu.

lines 192-193: "Viral samples were vortexed for 30 seconds prior to being pipetted to ensure the particles were evenly distributed through the suspension." I'm confused - shouldn't this lead to settlement of the particles rather than mixing? Please add a sentence of explanation to the text.

Reply: Virus particles would only settle if an ultracentrifuge was used with a much higher g force. We used a vortex mixer to agitate the suspension. This has been clarified.

lines 293-295: You write that biological INP play an important role in the terrestrial mid-latitudes. This is true not only there: Gong et al. (2020) showed that biogenic INP in the marine region of the Cape Verde, influenced by the Sahara, can contribute substantially to highly ice active INP, and that the majority of the INP there were supermicron in size. This, nevertheless, still leaves room for viruses to contribute. A brief discussion of this study in the context of your results could be added.

(Gong et al. (2020), Characterization of aerosol particles at Cape Verde close to sea and cloud level heights - Part 2: ice nucleating particles in air, cloud and seawater, Atmos. Chem. Phys., 20, 1451-1468, doi:10.5194/acp-20-1451-2020.)

Reply: The discussion has been amended to make it broader than just the mid-latitudes and the citation added.

line 486: Pietila -> Pielilä

Reply: Corrected to Pietilä, which is her (Maija K. Pietilä) correct name.

Table S4: Citations from this table should be added in the SI references. And it's "Ardon-Dryer"!

Reply: The name Ardon-Dryer is corrected in the table, and the references added.

Table S5: If you broaden the first column just a bit, it will look nicer ("nucleation" would be in one line, then).

Reply: Corrected.

---

## Author Comment (AC3)

**COMMENTS TO REFEREE 2**

This manuscript presents results on the ice nucleation activity (INA) of viruses. Viruses were not known to possess INA, so this manuscript reports novel and very interesting experiments on viral INA tested by immersion freezing over a range of temperatures between -15 and -35 degrees C. As the abundance of virus particles in the atmosphere is considered to be very high, their INA activity may be significant in climate processes. Therefore this manuscript falls within the scope of the journal.

I find these results interesting but not totally convincing.

The authors were careful to include important controls in their experiments: in addition to negative controls consisting of the buffer used for viral suspensions, controls included INA tests of the virus host strains, which is necessary to show that no carryover of bacterial INA was present in virus suspensions. The bacterial and archaeal host strains indeed did not possess any significant INA. Moreover, the authors examined the virus genomes and proteins for structural similarities of viral proteins to known bacterial proteins with INA. They found very little similarity.

However, an essential control to show no carryover of any bacterial INA into virus suspensions was not included in the experiments: an "extraction" control consisting of uninfected bacterial host cells going through the same procedure for extracting viruses. For example, this negative control would undergo the same centrifugation steps as for virus purification, and a sample of the gradient at the same depth as the virus band should be taken. Here is why I consider this essential:

All research on ice nucleation activity of bacteria was done at temperatures between -10 and 0 degrees centigrade. At this range, only intact bacterial cells are INA; purified bacterial INA proteins are significantly less active (at lower temperatures, attributed to the stabilization effect of the bacterial outer membrane). Other cell components after cell lysis are not active at these temperatures. The authors did their experiments of virus INA at significantly lower temperatures, from -15 to -35 degrees C. Indeed, at these temperatures the freezing temperature of most viruses tested is higher than the freezing temperature of buffer and of intact host cells. However, we do not know if bacterial cell lysates produced during virus extraction procedures are ice nucleation active at these temperatures. All we know from previous research of INA bacteria is that bacterial cell lysates are inactive at temperatures above -10 degrees C; we know nothing about their INA at -15 degrees and below. So we cannot rule out that the observed INA of viruses is in fact the activity of bacterial cell components carried over to virus suspensions. The authors partially addressed this concern in the experiment presented in Figure S3: a second purification of one virus preparation did not alter the INA activity of the virus. Did this additional purification eliminate any carryover of host cell components into the virus preparation? The conclusion would be more concrete if this was done for all viruses tested. An "extraction" control would clarify this.

Reply: The referee has picked up on a critical point, which we obviously have failed to discuss effectively. The referee is correct in saying that if a portion of the cell lysate were to make it

through the purification steps and were to nucleate ice, then this might be confused with an ice nucleation signal from virus particles. We consider this unlikely, but clearly did not discuss this important point well and have therefore improved this aspect of the paper.

The referee is incorrect in the statement that the ice nucleating activity of bacteria has not been tested at temperatures below -10°C. Wex et al. (2015), report measurements of ice nucleation by a lyophilized P. syringae to below -35°C. The P. syringae used by Wex et al. has very similar ice nucleating properties to other strains of ice-active P. syringae (Tarn et al., 2020) (Fig 6). Fig 11a from Wex et al. has been reproduced below and shows the cumulative number of ice nucleating sites per unit mass of P. syringae. This plot shows a steep increase in $n_m$ on cooling from 0 to about -10°C, but on further cooling to below -35°C no new ice-active sites become active (the line is flat). Snomax is produced in culture tanks, centrifuged, freeze dried and sterilized by radiation. In the experiments of Wex et al. the lyophilized bacteria was resuspended in pure water. Hence, some portion of the bacteria cells will lyse and the indication from the ice-nucleation data on P. syringae is that the bacterial cell lysates do not nucleate ice.

[Figure]

Figure 1. Ice active sites per unit mass of lyophilized P. syringae. From Wex et al. (2015).

Unfortunately, the 'extraction control' that the referee suggests is not possible because the virus infection causes the cells to lyse. Hence, removing the virus would not result in lysis and there would be no lysate. Trying to lyse the cells in some other way might create a lysate, but we would still be faced with the issue that the lysate may not have the same properties as the lysate from a viral mechanism.

For these reasons we chose to minimize the probability of unknown ice-nucleating components of the lysates by using the purification procedures. For Phi6, the purification protocols (the first

and the second one) have been optimized over decades and as it can be seen from figure S2, with optimized purification, the first and the second purification step lead both to high purity. As the referee points out, the fact that the second Phi6 purification step does not change the INA of the sample is a strong indication that it is the virus particles which nucleate ice. This, in addition to the lack of evidence that bacterial cells have any activity in this temperature range, suggests the observed INA was most likely due to the virus particles.

We have addressed this in Section 2.4 and now have a dedicated paragraph detailing these points. In addition, we have added a brief paragraph to the beginning of the Methods section to stress the lengths which we went through to purify the virus particles.

Specific comments on the manuscript:

Page 2, line 52: please include the original reference on bacterial ice nucleation from S.E. Lindow et al (LINDOW, S.E.;ARNY, D.C.;UPPER, C.D.;BARCHET, W.R. The role of bacterial ice nuclei in frost injury to sensitive plants. In: Li, P.H.; Sakai, A. (eds.). Plant cold hardiness and freezing stress: Mechanisms and crop implications. Academic Press. New York. 1978. p. 249-263.).

Reply: The correct reference (Lindow et al., 1972) has been added).

Page 5, Section 2.1 of Methods: The suspension of archaeal virus particles was done in K-phosphate buffer with NaCl. This information is missing from this section and the reader does not understand the use of a NaCl negative control in INA experiments in section 2.3 (line 204) later in the manuscript.

Reply: Thank you for the comment. The following explanation was added to lines 209-211: "NaCl (500 mM) was added to the buffer for the archaeal cells and viruses due to them being classified as extreme halophiles that require NaCl for optimum growth or infectivity."

Page 6, line 156 and Figure S2: Are these viral protein profiles in agreement with reports in literature? Do they confirm the absence of non-viral proteins originating from host bacteria?

Reply: Yes, the proteins are known and similar than reported in previous research. The structural proteins of Phi6 (Figure S2) have been determined by N-terminal sequencing confirming that they are from viral origin (Mindich et al., 1988).

Page 7, Section 2.3: Please provide more details about the experimental procedure: mention the range of temperatures used during the ice nucleation experiments; how many droplets from each sample were tested; what was the rate of temperature drop during the tests.

Reply: These details have been added.

Page 8, line235: why the most ice nucleation active virus Phi12 (shown in Fig. 1E, page 21) were not included in the study of the sub-viral particles' INA?

Reply: Dissociation into subviral particles is a laborious process which takes years to optimize and succeed. In this study, we used virus dissociation experiments to the two viruses (Phi6 and PRD1) to which those experimental protocols were available.

Page 9, lines 246-247: please rephrase this sentence with respect to syntax. Excessive use of word "hence".

Reply: Corrected.

Page 9, section 3.2 title: modify to "Genetic analysis of ice nucleation active virus particles".

Reply: Corrected.

Page 9, line 273: Please rephrase: "...the source of INA might also be of proteinaceous origin".

Reply: Corrected.

Page 10, line 274: Please rephrase: "virus particles possessing similar structure and function to known bacterial ice nucleation proteins as explanation….".

Reply: Corrected.

Page 10, lines 275-276: Please improve this sentence.

Reply: The previous sentence: "This hypothesis was approached using ice nucleation motifs, specific and conserved short sequences in IN proteins, to search for the proteins potentially capable of nucleating ice.", noted by the referee, was corrected to "Proteins with potential ice nucleation activity were screened by searching for conserved short sequences called ice nucleation motifs in the amino acid sequence." .

Page 11, lines 305-316: Please discuss whether the reported marine organics and sea spray aerosols are chemically or physically related somehow to viruses (information from the 3 references mentioned in this paragraph).

We have added the following text at this point: 'Tests indicate that these INP are sensitive to heat and are also pass through 200 nm filters (Wilson et al. 2015; Schnell, 1975) and while it is unclear exactly which component of the sea water nucleate ice it has been pointed out that virus particles are of the right size (Wilson et al. 2015).'

Figures S5 and S7: the archaeal hosts have some INA too, higher than the buffer and close to or same as the respective viruses. This was not the case with the bacterial hosts: they all had the same INA as the buffer controls. Therefore the conclusion that the INA of the viruses in these figures is not some "contamination" effect by the host is not totally supported.

Reply: It is correct that in Fig S5 the host cells nucleate ice at a similar temperature to the virus, hence we report the INA activity in Fig 1 as a limiting value (open symbols). We have added the

following statement to the Fig 5 caption to clarify: '(hence we report the INA of HRPV6 as a limiting value in Fig. 1). For Fig 7, even though there is some activity, the activity of the virus is greater. Also, bear in mind the purification step removes the host cell material.

References

Tarn, M. D., Sikora, S. N. F., Porter, G. C. E., Wyld, B. V., Alayof, M., Reicher, N., Harrison, A. D., Rudich, Y., Shim, J.-u., and Murray, B. J.: On-chip analysis of atmospheric ice-nucleating particles in continuous flow, Lab on a Chip, 10.1039/D0LC00251H, 2020.
Wex, H., Augustin-Bauditz, S., Boose, Y., Budke, C., Curtius, J., Diehl, K., Dreyer, A., Frank, F., Hartmann, S., Hiranuma, N., Jantsch, E., Kanji, Z. A., Kiselev, A., Koop, T., Mohler, O., Niedermeier, D., Nillius, B., Rosch, M., Rose, D., Schmidt, C., Steinke, I., and Stratmann, F.: Intercomparing different devices for the investigation of ice nucleating particles using Snomax (R) as test substance, Atmos. Chem. Phys., 15, 1463-1485, 10.5194/acp-15-1463-2015, 2015.

---

## Referee Report (RR1)

The authors describe the virus preparation protocols and purification methods in a much more detailed manner in the revised Section 2.1 of Materials and Methods. The authors disputed my recommendation for a bacterial lysis control to be included in their experiments in their reply to my comments, stating that Snomax, used in a 2015 study as a test ice nucleation active substance to compare various instruments measuring ice nucleation particles, also contained lysed cell debris; so my comment about the potential ice nucleation activity of bacterial lysates was indirectly addressed then. I do not fully agree with this, as cell lysis of Pseudomonas syringae during Snomax preparation is not the same as lysis by a virus, just as bacterial cell debris after lysis with any other method would also be different from the cell debris after lysis by a virus. I agree with their comment about the difficulty of obtaining a proper control of bacterial lysates. The critical point here is to assure the purity of virus preparations from bacterial cell debris, and this is addressed properly in the revised section 2.1 of Materials and Methods, describing the production and purification of virus particles free from bacterial cell debris.

All my other comments were fully addressed. Some very recent references were also added. After this revision, the manuscript is greatly improved. The authors report novel and very significant findings on the ice nucleation activity of viruses, and I am happy to recommend this manuscript for publication in this journal.